# Hyaluronic Acid: Known for Almost a Century, but Still in Vogue

**DOI:** 10.3390/pharmaceutics14040838

**Published:** 2022-04-11

**Authors:** Anna Lierova, Jitka Kasparova, Alzbeta Filipova, Jana Cizkova, Lenka Pekarova, Lucie Korecka, Nikola Mannova, Zuzana Bilkova, Zuzana Sinkorova

**Affiliations:** 1Department of Radiobiology, Faculty of Military Health Sciences, University of Defence, 500 01 Hradec Kralove, Czech Republic; alzbeta.filipova@unob.cz (A.F.); jana.cizkova@unob.cz (J.C.); lenka.pekarova@student.upce.cz (L.P.); zuzana.sinkorova@unob.cz (Z.S.); 2Department of Biological and Biochemical Sciences, Faculty of Chemical Technology, University of Pardubice, 532 10 Pardubice, Czech Republic; jita.kasparova@gmail.com (J.K.); lucie.korecka@upce.cz (L.K.); nikola.mannova@upce.cz (N.M.); zuzana.bilkova@upce.cz (Z.B.)

**Keywords:** hyaluronic acid, radiation, hyaluronan receptor, therapeutic application, radioprotection

## Abstract

Hyaluronic acid (HA) has a special position among glycosaminoglycans. As a major component of the extracellular matrix (ECM). This simple, unbranched polysaccharide is involved in the regulation of various biological cell processes, whether under physiological conditions or in cases of cell damage. This review summarizes the history of this molecule’s study, its distinctive metabolic pathway in the body, its unique properties, and current information regarding its interaction partners. Our main goal, however, is to intensively investigate whether this relatively simple polymer may find applications in protecting against ionizing radiation (IR) or for therapy in cases of radiation-induced damage. After exposure to IR, acute and belated damage develops in each tissue depending upon the dose received and the cellular composition of a given organ. A common feature of all organ damage is a distinct change in composition and structure of the ECM. In particular, the important role of HA was shown in lung tissue and the variability of this flexible molecule in the complex mechanism of radiation-induced lung injuries. Moreover, HA is also involved in intermediating cell behavior during morphogenesis and in tissue repair during inflammation, injury, and would healing. The possibility of using the HA polymer to affect or treat radiation tissue damage may point to the missing gaps in the responsible mechanisms in the onset of this disease. Therefore, in this article, we will also focus on obtaining answers from current knowledge and the results of studies as to whether hyaluronic acid can also find application in radiation science.

## 1. Introducing Hyaluronic Acid

The importance of hyaluronic acid (HA) has increased over the past 10 years due to new biomedical applications exploiting its full biocompatibility and unique bioreactivity. HA is a polysaccharide belonging to the group of glycosaminoglycans (GAGs), which is a group of highly sulfonated, complex, linear polysaccharides manifesting a number of important biological roles. HA is in several respects an exception among GAGs. Within the cell, GAGs are synthesized in the Golgi apparatus network and subsequently bind covalently to proteins while proteoglycans are being created. Based on their distinct, repeating disaccharide units, GAGs can be divided into four major groups: heparin/heparan sulfate, chondroitin sulfate/dermatan sulfate, keratan sulfate, and hyaluronan. These polymers serve as cell surface molecules or in extracellular matrix (ECM) [1,2,3,4].

Hyaluronic acid (also known as sodium hyaluronate or hyaluronan) is a straight-chain, natural polysaccharide and the only nonsulfated GAG composed of alternating (1–4)-β d-glucuronic and (1–3)-β N-acetyl-d-glucosamine units [5]. Both carbohydrate units are spatially related to glucose; therefore, in the β-configuration, it is possible for all their bulky groups (hydroxyl and carboxyl groups and the anomeric carbon on the neighboring sugar) to be in sterically favorable planes, while all the small hydrogen atoms occupy less sterically favorable axial positions [6]. This chemical structure of HA (Figure 1) is energetically very stable because of interactions between hydrophobic and intermolecular hydrogen bonds and the acetamide and carboxylate groups [7].

HA was first isolated from the bovine eye vitreous body by Karl Meyer and John Palmer in 1934 [8]. The name “hyaluronic acid” was created as a conjugation of the two words: hyaloid (vitreous body) and uronic acid. Although Meyer and Palmer are generally considered to be the discoverers of HA, the very first mention of the HA molecule dates back to 1918, when Levene and Lopez-Suarez isolated an unknown polysaccharide from a vitreous body and umbilical cord blood that they called mucoitin sulfuric acid [9]. The precise chemical structure of HA was described after almost 20 years of research also by Meyer’s group [10].

Hyaluronan is one of the most hygroscopic molecules found in nature [11]. The molecular chains are intertwined to form highly viscous and elastic solutions, even at very low HA concentrations. This phenomenon can be observed in solutions containing as little as 1 μg/mL HA, which is one of the reasons for this molecule’s unique rheological properties [12]. The strong hydrophilicity of HA is, therefore, the physical basis for its widespread presence in the body. HA is found in the ECM of all vertebrates’ tissues [13], but it is also as a component of an extracellular capsule produced naturally by various species of bacteria [14]. In the human body and especially in connective and soft tissues, HA exists primarily in the form of high molecular weight chains (HMW-HA) of greater than 10 kDa [15]. HA hydrophilicity is the fundamental and a very important property for the control of tissue hydration and osmotic balance [16]. Due to its strong anionic nature, HA can capture water molecules with winding chains, leading to an important physicochemical property: water retention. The large number of hydroxyl groups greatly contributes to this property [17]. HA can retain water up to 1000 times its own weight [18]. The water content increases with rising relative humidity [19]. Hydration parameters are, nevertheless, independent of HA’s molecular weight [20].

One of the most controversial capabilities of hyaluronan in the body is its antioxidant activity as an effective “scavenger” of free radicals. It is well known that reactive oxygen species (ROS) can lead to oxidative stress, which implies a change in intracellular redox homeostasis and leads to a serious imbalance between the production of reactive species and antioxidant defense [21]. On the one hand, ROS degrades HA [22]; on the other, the antioxidant capacity of hyaluronic acid is scientifically proven [23]. Many years of intensive research and numerous published scientific papers have been needed to clarify this incongruity. More precise mechanisms will be explained in the next section, because this feature is critical for the intention of our review.

Although HA is not branched and contains no sulfate groups and despite its relatively simple structure, it intermediates numerous very important molecular functions, including intracellular signaling. This review in the next sections will describe metabolism from synthesis and physiological occurrence to enzymatic and radical degradation (Section 2) and the properties of different chain size of HA (Section 3). The unique biological functions of HA are largely attributed to its specific binding and interaction with its specific proteins—hyaladherins—introduced in next section [24]. The purpose of this review is to focus on the potential applications of HA in radiation biology and radioprotection as well as to summarize the most important functions that contribute to structural and physiological properties of tissues. Ionizing radiation (IR) has direct and indirect effects on living cells. The direct effect is caused by the absorption of radiation energy and breaking of molecular bonds in cells. Much of the subsequent damage caused by IR is due to its indirect effects as a result of ROS generation. The damage resulting from radiation is a loss of specialized cells, manifesting itself in subsequent disruption of physiological functions due to changes in tissue organization [25]. Furthermore, this review summarized the unique linking between HA, ionizing radiation, and radiobiology, which have not been conducted yet. From this perspective, the HA metabolic pathway may be a very attractive, novel, and suitable target not only for its contribution to mitigating damages but also in radioprotection. A view of these scientific fields is not contemporaneously available, and a review of these fields allows us to complete the missing knowledge gap of enhanced HA possibilities.

## 2. A Unique Metabolism from Beginning to End

HA is uniquely synthesized on the inner side of the plasma membrane by membrane-bound glycosyltransferases. From a molecular biology perspective, these HA glycosyltransferases disproved the molecular dogma that “one enzyme carries only one sugar group” [26]. The key year for this discovery was 1993, when DeAngelis et al. for the first time successfully identified, sequenced, and cloned hyaluronan synthase A (hasA), a gene from *Streptococcus pyogenes* [27,28]. A second turning point came in 1997, when a second group of HAS genes was identified in bacteria *Pasteurella multocida* (PmHAS), also by a team led by DeAngelis [29]. Although this is the only PmHAS of the latter group known to date, chondroitin synthase proteins that are structurally similar have been identified and cloned in two bacteria: *P. multocida* and *Escherichia coli* [30]. Since that time, more than 20 eukaryotic, bacterial, and even viral genes or cDNA sequences have been described for HAS [30]. Based on the more advanced technology and methods used in genome identification and sequencing, whether prokaryotic or eukaryotic, and the amount of experimental results obtained, some authors have already suggested that it would be appropriate to classify HAS into three groups [31]. It is beyond the scope of this review to explain or even mention all the basic information concerning the individual species of HAS, but this topic is described in great detail in several excellent reviews (such as DeAngelis [26], Weigel [32], and Siiskonen et al. [33]). Within the scope of our article, we will only briefly mention HAS in mammalian species, basic information, and topics that are currently of great and particular interest. 

In vertebrae species, three isoforms of HA synthase (HAS1–3) have been identified (except that the genus of Xenopus laevis has 4 genes [34]) that are specifically expressed in various time intervals and tissues under different physiological or pathological conditions [35]. These are evolutionarily quite highly conserved proteins, sharing 50–71% of amino acid sequences, but the gene sequence of each isoform is localized on a different chromosome (HAS1—hCh19; HAS2—hCh8; HAS3—hCh16) [36,37]. In general, HAS1 synthesizes HMW-HA with masses 2 × 10^5^ to 2 × 10^6^ Da and produces chains of variable lengths, but is the least active isoform. In contrast, HAS3 synthesizes mainly HA polymers of low molecular weight (LMW-HA; 1 × 10^5^ to 1 × 10^6^ Da), but it is the most active of all isoforms. Chains produced by the HAS2 isoform are also high molecular weight HA, similarly as are those of HAS1, but HAS2 tends to produce more uniform chains with masses greater than 2 × 10^6^ Da [38,39,40]. In contrast with the physiological properties of individual HAS enzymes, in the work of Itan et al., with recombinant proteins of individual mammalian HAS enzymes, the highest Michaelis constant (KM) value was reached by HAS1, which may cause the lower synthetic rate of nucleotides [41]. This study also suggests that the concentrations of individual nucleotide precursors, as well as the overall nature and intrinsic properties of individual synthases, are important factors in the regulation of HA chain creation. One of the most critical discoveries at the turn of the millennium in hyaluronan science was clarification as to the roles of these synthases in the stages of embryogenesis and, particularly, those of HAS2. Camenisch et al. found that HAS-deficient mice embryos (HAS2^−/−^) have severely impaired cardiac and vascular morphogenesis and develop highly severe or lethal defects [42]. Nevertheless, HAS1^−/−^ and HAS3^−/−^ mice, as well as double knockouts, are viable and fertile [43]. Of particular importance is mainly HAS2, which is the most abundant isoform in adult tissues and is involved in the processes of tissue development, repair, growth, and regeneration in the cases of insult. For these reasons, the expression and impact of the HAS2 isoform also will be important to the aims of our review and will be discussed further in the next section.

Each HAS enzyme is capable of de novo chain synthesis (Figure 2), and the differences are the molecular weights of the chains being created and, hence, in their biological functions [44]. HAS uses cytosolic UDP-N-acetylglucosamine and UDP-glucuronic acids (UDP-GlcNAc and UDP-GlcUA) as precursors to polymerize the HA chain without the need for a primer, anchor protein, or lipid [45]. The amounts of nucleotide precursors and their relative proportions constitute one of the major limiting factors of the HA chain, and the results from mass spectrometry analyses confirm that HAS1 produces also chitin oligomers in the presence of UDP-GlcNAc and absence of UDP-GlcUA [46]. The number of repeating disaccharides on a completed hyaluronan molecule may reach 10,000 or more disaccharide units, and the molecular weight can grow to as great as 4 × 10^6^ Da. The average length of a disaccharide is approximately 1 nm. Thus, a hyaluronan molecule of 10,000 repeating units could elongate by 10 μm, which is approximately the diameter of a human erythrocyte [47]. After synthesis, HA is extruded into extracellular space without any need for transport or exocytosis.

Intensive research on HA-related enzymatic structures is still ongoing today because the precise mechanisms of their translocation to cell membrane, the regulation of the molecular transcription mechanism and post-transcriptional modifications, and even the complete physiological roles of individual HAS enzymes are not yet known [48]. The expression of *Has* genes is also significantly influenced by the increased activity of growth factors and cytokines or signaling molecules under certain conditions in tissues [49]. The work of Tlapak-Simmons et al. was the first to unambiguously determine the direction of HA synthesis in individual groups by confirming that class I HAS enzymes polymerize chains at the reducing end [50]. The current capabilities for co-immunoprecipitation and Förster resonance energy transfer (FRET) analysis are rewriting and adding to the knowledge about HAS functions and actions. Deen et al. identified the Rab10 protein as the first suppressor for trafficking to the plasma membrane, which is responsible for controlling HAS3 levels [51]. Karousou and her research team confirmed that HAS2 is regulatory ubiquitinated [52], which affects activity and stability and further demonstrates that HAS2 is capable of dimerization. The work of Bart et al., however, has shown that all HAS isoenzymes form homomeric and heteromeric complexes with one another. They even have suggested that the homomeric and heteromeric interactions detected among the different HAS isoenzymes are specific and potentially influence the enzymatic activity of native HAS proteins, perhaps in a dominant negative manner [53].

HA is a ubiquitous component of the ECM and pericellular space, evolutionarily occurring already from bacteria, algae, and mollusks; some viruses; and to mammalians from among chordates, including humans [54]. It has not yet been observed, however, in fungi, plants, or insects (although insect venom contains enzymes cleaving HA (hyaluronidases) [55,56]. The fruit fly Drosophila melanogaster, however, can produce HA in vivo after transfection with the mouse HAS2 gene, albeit with significant morphological defects due to the bulk accumulation of HA in the extracellular space [57]. In an adult human weighing 70 kg, the amount of HA present is approximately 15 g. More than one-half of this amount is present in the skin (primarily in the intercellular space of the dermis), in the synovial fluid of the knee joint there is approximately 2–3 mg/mL, and the vitreous body of the eye contains approximately 200 µg/mL [58,59]. Moreover, the umbilical cord contains a significant amount of HMW-HA (up to 4 mg/mL) [10]. HA is nevertheless present in all body tissues and fluids, for instance, in lungs (15–150 mg/g), where it is mainly localized in the peribronchial and interalveolar space [60], in skeletal muscles [61], as well as in lymph. Surprisingly, a lymph has been found to contain primarily LMW-HA and serves as its pool [62].

In the body, HA occurs in many different forms, including circulating [63], bound by various HA-binding proteins (known as hyaladherins) [64], associated with tissue or ECM [65], or electrostatically or covalently bound to other matrix molecules [66]. HA does not bind directly by covalent, chemical bonds to the protein core; therefore, it does not form proteoglycans [67], but it is known that it can form aggregates with proteoglycans occurring in ECM [68,69].

Moreover, the degradation of HA is permanent, leading to extremely rapid turnover of HA in the body. HA’s transformation times range from a few minutes (in the bloodstream) to 2 days, but a maximum of up to 70 days in the vitreous of the eye [70]. Degradation takes place at different locations, with approximately 30% being degraded locally in tissues and the remaining 70% entering the lymphatic drainage [71]. Degradation in vivo can take place via two main and simultaneous mechanisms: enzymatic degradation and chemical depolymerization [72]. The enzymatic degradation of HA occurs via the endoglycosidase family of enzymes, consisting of homologous proteins generally referred to as hyaluronidases (HYALs) and specifically hydrolyzing the β-1,4 linkage predominantly in HA molecules [73]. Recently, it has been confirmed that HYALs have limited ability to slowly degrade other GAGs, namely chondroitin and chondroitin sulfates [74]. This group of enzymes was also classified by Meyer in 1971 [75]. Based upon this exceptionally precise classification, HYALs are divided into three classes with different cleavage end products: bacterial lyases (EC 4.2.99.1) [76]; endo-β-glucuronidases occurring in leeches and bark beetles (EC 3.2.1.36) [77]; and those exceptionally cleaving the β-1,3 bond and mammalian hydrolases (EC 3.2.1.35) [78,79]. In humans, six gene sequences for HYALs have been identified to date: hyaluronidases 1–3 (HYAL1–3) are localized on hCh3p21.3, while HYAL4, PH20/SPAM, and HYALP1 pseudogenes are clustered on hCh7q31.3 [73]. The expression of individual HYALs is tissue-specific [80]. We will briefly mention, at least, the basic information about the individual HYALs.

The first to be isolated and purified was HYAL1 [81], a 57-kDa polypeptide glycoprotein that is the major degradative enzyme present in serum [82], a cell cycle regulator, and a candidate tumor suppressor gene [83]. HYAL2 is a glycosylphosphatidylinositol (GPI)-linked enzyme with low activity attached to the outside of the plasma membrane; it cooperates with HA receptors present on the cell surface and primarily with a cluster of differentiation 44 (CD44) [84]. Both HYAL1 and HYAL2 are highly expressed in human somatic tissues and are optimally active at acidic pH [85]. HYAL1 works together and sequentially with HYAL2 to degrade HA. HYAL2 initiates the degradation of HMW-HA chain into fragments of up to 20 kDa (approximately 50–60 disaccharide units), which are transported first to endosomes and then to lysosomes [86], where HYAL1 steps in and by its action, they are degraded into tetrasaccharides of 800 Da, a predominant end-product of HYALs degradation [87]. The resulting oligosaccharides become substrates for the terminal reaction of two lysosomal β-exoglycosidases: for D-glucuronidase, which hydrolyzes terminal nonreducing glucuronic acid, and for p-N-acetyl-D-hexosaminidase, hydrolyzing nonreducing UDP-GlcNAc [78]. The catabolic pathway of HA generates different size fragments, which widely differ in their biological HA properties. HA catabolism is a highly ordered, carefully controlled process under the regulation of the individual enzyme activities. Due to that, the LMW-HA chains are very active inflammatory biological molecules; actually, tetrasaccharides produced by HYAL1 can be recognized as “dangerous”. It also has been found that hyaluronan depositions and turnovers are even more intense and rapid in patients’ tumor tissues with a larger proportion of LMW-HA [88]. Biological properties of different HA chain are described in next section, but due to this biological divergence, a precise control mechanism must exist. The inhibitors of HYALs are ubiquitously present in every tissue, but this class of molecules is still not fully known, and their mechanism remains undiscovered [89,90].

Interestingly, HYAL4 and PH20/SPAM also are GPI-anchored proteins. PH20/SPAM1 is the most active, multifunctional GPI-linked sperm protein important for fertilization. It is localized on the anterior surface of sperm that facilitates the surrounding the oocyte. It also degrades HA chains from oligo- to tetrasaccharides. Testicular tissue is a physiological source of bioactive hyaluronidase [91]. Although currently available evidence indicates that HYAL4 has no degradation activity against HA, it predominantly and uniquely degrades chondroitin sulfate [92]. The latest discovery is hyaluronidase protein Transmembrane Protein 2 (TMEM2) located on the cell’s surface and is strongly active in the pH of the extracellular environment. This protein does not share structural homology with other HYALs, but it has 48% amino acid homology with cell migration-inducing proteins (CEMIP; discussed below). Neither the identification of TMEM2 nor its role in HA catabolism has been completely resolved [93]. Not much is known, either, about the activity and functions of individual HYALs. Such information as is currently available and only fuels more new questions and the need for intensive research. How is it possible that HYAL1 is the only protein present in the circulation and at the same time a degradative lysosomal enzyme in a pH 4.5 environment? HYAL3 also is a mystery because, despite its widespread expression in various tissues but very low activity, it cannot be identified by available hyaluronidase assays [94]. The clarification of these questions will lead to a more detailed understanding of the degradation pathway but also of the overall HA metabolism and signaling pathway, implying new, more advanced modulation strategies in situations of tissue damage or tumor disease.

The second method of degrading HA, chemical depolymerization, is intermediated through reaction with ROS, including superoxide, hydrogen peroxide, nitric oxide, hypohalous acids, and hydroxyl and peroxynitrite radicals at the glycosidic bond [95]. The result is the subsequent fragmentation of the polymer into various LMW-HA chains up to the size of oligomers. Free radicals and HYALs may coordinate their activities in certain pathological situations, but the relative roles of the different mechanisms are not known [96]. The depolymerization of HA will be discussed in detail in the next section, because this is a direct consequence of IR on the body and a product of this reaction is the significant disturbance of the equilibrium at the level of physical, chemical, and especially biological homoeostasis of the extracellular matrix. With its versatile properties, such as its biocompatibility, non-immunogenicity, biodegradability, and viscoelasticity, HA is an ideal biomaterial finding wide application, particularly in the cosmetic industry and in non-surgical aesthetic dermatology [97] and ophthalmology [98]. In the new millennium, it also has become a very important molecule in tissue engineering [99], drug-delivery systems [100], and cancer therapy [101].

## 3. HA Fits Every Size

In homeostasis, HA in its HMW-HA form is found in almost all human tissues. Molecular weight and the synthesis/degradation ratio are the key factors defining the biological effects of HA and having significant impacts on its physiological functions and activities. HA polymers occur in the body in a variety of sizes and have an enormous number of biological functions that are often contradictory and opposite [102]. Recent scientific work points to the need to divide HA chains according to their size into more groups than only HMW-HA and LMW-HA, because different chain sizes also exhibit different properties and even tetraoligomers are biologically unique [103].

The most common, physiological, endogenous, and native form of HMW-HA is as a molecule with a mass above 1000 kDa. The average molecular weight is generally greater than 2000 kDa (2 × 10^3^–10^5^ sugars and more), usually in the range of 6000–8000 kDa [104]. HA polymers are the largest molecules occurring in ECM. In general, an HA molecule of this size is a “cellular bodyguard to protect homeostasis.” HMW-HA has exceptional biophysical properties. It serves as a hydrating tissue lubricant [105]; is strongly anti-angiogenic [106] due to its inhibition of gene expression, proliferation, and migration of endothelial cells [107]; is strongly anti-inflammatory [108]; forms a matrix within which many different anti-inflammatory chemokines, cytokines, and growth factors are produced in many cell types [109]; acts as a potent signaling molecule upon binding to multiple receptors [110,111]; and, in addition, is involved in the wound healing process [112]. HMW-HA is also immunosuppressive because of its ability to coat the cell’s surface, thereby preventing the access of ligand to surface receptors [113].

Several papers have confirmed that exogenously administered HMW-HA also prevents liver injury by reducing proinflammatory cytokines in an in vivo model of T cell-mediated injury [114], as well as protects lung tissue from damage (found in a model of lipopolysaccharide-induced lung injury, with HA = 1600 kDa) [115]. In lungs, hyaluronan can protect alveolar epithelial cells from apoptosis by means of nuclear factor kappa light chain enhancer of activated B cells (NF-κB; the preparation Healon, from Kabi Pharmacia, Sweden, was administered) [116]. The antioxidant properties of added HA (2700 kDa) to ex vivo cultured bovine cartilage slices were also proven, in which case ROS synthesis caused by mechanical stress was nullified [117]. HMW-HA concentrations in plasma can increase many fold upon certain injuries or damage to the body. Under pathological conditions such as inflammation [118] or tumorigenesis (nicely summarized in an article by R. Stern [119]), there is increased enzymatic fragmentation of HA as a result of *Has* gene expression, as well as nonenzymatic fragmentation leading to increased amounts of LMW-HA oligomers [120,121,122].

Although the definition of HMW-HA depending upon its size and biological effects is fairly straightforward, there is a problem with defining LMW-HA across the available literature because not every study reports the size of molecules that are labelled as “short-fragment HA”, “low molecular weight HA”, or “HA oligosaccharides”. In the work of Cyphert et al., for example, anything between 20 monosaccharides in length and 1 million Da is referred to as LMW-HA [123]. On the other hand, Monslow et al. categorized HA already into four groups, defining LMW-HA as chains of size 10–250 kD, and introduced the term intermediate ([medium] MMW-HA) referring to size 250–1000 kDa [124]. Depending upon the cell type, the different HA chain lengths influence the formation of the receptor complex and, thus, also the resulting activation and/or regulation of the signaling cascade. In the future, it will be necessary always to define the size of the molecule and, thus, enable faster progress in discovering biological effects of individual HA fragments. Rayahin et al. confirmed unequivocally in their work that HA chains of four different lengths—5 kDa, 60 kDa, 800 kDa, and 3000 kDa (identical to the classification by Monslow et al. [124])—differed significantly in the resulting signaling as well as to the final activation of individual macrophage phenotypes [125].

In general, we can state that chains shorter than 10^6^ Da are distinctive bioactive molecules: angiogenic [126] and inflammatory [127]. HA fragments shorter than 400 kDa are recognized in the body as danger-associated molecular patterns (DAMPs) [128]. Fragment size is important in signaling activation. Fragments larger than 500 kDa or smaller than hexamers already contribute strongly as amplifiers of inflammation through interaction with Toll-like receptors 2 (TLR2) [129] or 4 (TLR4) [130]. The interaction of LMW-HA with TLRs promotes mitogen-activated protein kinase (MAPK) phosphorylation and NF-κB translocation to the nucleus [131]. A fascinating thing about this molecule is that nearly any chain length, even a very small one, can activate some kind of cellular action in the body. In a study with hyaluronan dodecasaccharides, the induction of endothelial morphogenesis via the upregulation of the gene for chemokine CXCL1/GRO1 via binding to cluster determinant 44 (CD44) was demonstrated [132]. Indeed, HA-specific hexasaccharides activate the transcription factors of chondrogenesis, especially metalloproteinase-3 (MMP-3) and type II collagen [133,134]. Hyaluronan tetraoligomers are inducers of heat shock proteins [135]. Based upon the findings reported in scientific papers, it can be argued that the macrophage population is significantly modulated by HA’s action. Macrophages constitute an essential cellular component of the inflammatory response. The difference in chain size influences significantly the capability for phagocytosis [136], the induction of expression of various cytokine and chemokine genes [137,138,139], and nitric oxide production [140,141]. Moreover, dendritic cells (DCs) are activated by the presence of short HA fragments [130,142], and the resulting molecular weight of HA is also important for interactions with CD44 [143]. Although a more detailed description of the biological properties and effects of short chains is far beyond the scope of this article, we can recommend a number of excellent publications wherein the subject is described in great detail, such as Stren et al. [144], Powell et al. [145], Misra et al. [146], and Bohaumilitzky et al. [147].

A fascinating natural curiosity is the animal species known as the naked mole rat (Heterocephalus glaber). In this animal, we can find a unique HA with a very high molecular weight of more than 12 million Da [148]. The species also has an unusually long life (about 10 times longer than other Rodentia orders) and an incredible (but not absolute) resistance to tumorigenesis [149]. Recent findings provide evidence of a link between cancer resistance and its extreme HMW-HA [150]. Perhaps, this unattractively looking creature and its extremely long HA will be the key to curing cancer and to eternal life. However, much more research focusing on functional diversity and the biology of HA is needed here. Much more likely than eternal life is that the clarification of all molecular mechanisms and signaling pathways will greatly facilitate the development of new HA-based therapies in the future.

## 4. Old and New Partners in Action

As already mentioned, HA in vivo can not only act as a proinflammatory but also an anti-inflammatory molecule, and the resulting molecular action depending upon the binding of a polymer of a given length and the presence of a specific binding protein partner—hyaladherin (HABP)—is summarized in Table 1 [151,152]. Hyaladherins make up a heterogeneous group of diverse proteins with the ability to bind HA and are primarily localized either extracellularly as receptors on the cell surface or intracellularly [153]. These proteins can be grouped according to the interaction domain present. One group of interactions with HA is mediated through a protein domain called the Link module, also known as the proteoglycan tandem repeat, with the size of approximately 100 amino acids [154]. The Link module is a characteristic consensus sequence containing four disulfide-linked cysteines [155]. This sequence is abundant in G1-domains among the proteoglycan molecules within the ECM, aggrecan [156], versican [157], neurocan [158], and brevican [159], collectively termed “hyalectans” or “lecticans”. HA forms huge non-covalent complexes with these extracellular proteins providing load-bearing functions in articular cartilage and elasticity to blood vessels, as well as contributing to the structural integrity of many tissues such as skin and brain [160,161]. The Link module superfamily includes also surface receptors CD44 [162], lymphatic vessel endothelial hyaluronan receptor 1 (LYVE-1) [163], hyaluronan receptor for endocytosis (HARE)/stabilin-2 [164], and tumor necrosis factor-stimulated gene-6 (TSG-6), where the Link module was initially determined [165]. However, a number of proteins that interact with HA lack this module. The second interaction motif identified as having high affinity and specificity was a 9-amino-acid basic B(X7)B motif [166]. This sequence is found in a group of proteins comprising hyaladherins: the receptor for hyaluronan mediated motility (RHAMM) [167], essential cell cycle regulatory factor CDC37 [168], inter-α-inhibitor (IαI) [169], and CD38 [170]. Other motifs capable of binding HA, such as short base sequences identified on pigment epithelium-derived factor (PEDF) [171] or peptides containing the sequence Arg–Arg [172], have been identified. Nevertheless, their exact role in relation to HA is not yet precisely known. HABPs generally interact with at least 6–10 HA oligosaccharides, but a single HMW-HA chain may bind 1000 protein molecules [173]. It is important to note that the association of HA with HYALs changes dynamically in times of tissue damage, inflammation, and organ development, whereby a great plasticity and the versatility of the HA biological properties were achieved [174,175]. Moreover, HYALs have important biological functions beyond their bonds to HA and, thus, contribute significantly to expanding the range of activities involved in HA signaling within the body.

The first mentioned and best characterized of all HABPs is CD44, also known as phagocytic glycoprotein-1 (Pgp-1). It is a single-pass, type I transmembrane glycoprotein expressed by a wide variety of hematopoietic and non-hematopoietic cells [176,195,196]. CD44 is ubiquitous throughout the body and has a molecular weight of 85–200 kDa [197]. CD44 is encoded by a single gene located on human chromosome 11 or mouse chromosome 2 [198,199]. Because of alternative exon splicing, the gene contains 20 exons and multiple variants of CD44 can be identified in the body. The first five and last five exons are constant and exons 6 to 15 are variable [200]. The smallest standard CD44 isoform (CD44S) is ubiquitously expressed mainly on leukocytes, while variant isoforms (CD44v1–v10) are expressed in several epithelial tissues and larger ones during tumor progression [201,202]. Recently, CD44 was also identified as a typical surface marker for cancer stem cells, being a critical regulator of cancer stemness, self-renewal, tumor initiation, and metastases [203]. A link with the expression of one or more CD44v in individual types of cancer has been proven, for example in: breast cancer (CD44v3,8–10) [204,205], head and neck squamous carcinomas (CD44v3) [206,207], prostate cancer (CD44v6) [208], pancreatic cancer (CD44v4,5,6,9) [209,210,211], and gastrointestinal cancer (CD44v6) [212,213]. Therefore, CD44v expression is currently being studied intensively, as individual isoforms may serve as valuable diagnostic or prognostic markers, as well as ideal targets for developing clinical therapeutics in many types of cancer [214]. Also, the data obtained suggest that CD44 expression and signaling contribute to regulation of tumor radioresistance response. This mechanism will be explained in the next section. The main ligand of the CD44 receptor is HA, but it can interact with several other molecules, such as osteopontin [215], matrix metalloproteases (MMPs) [216], fibronectin, fibrinogen, and selectins [217], and many others. HA/CD44 interactions activate many signaling pathways of protein kinases, cytoskeletal changes, as well as intracellular pathways including Ras, MAPK, and phosphoinositide 3-kinases (PI3K) that contribute to cancer cell division, proliferation, invasion, and angiogenesis, as well as metabolic shift [197,198,202,218,219]. Lesley et al. identified three states of CD44 activation by HA: active CD44, which constitutively binds HA; inducible CD44, which does not bind HA or binds it only weakly unless activated by inducing factors (e.g., monoclonal antibodies, cytokines); and inactive CD44, which does not bind HA even in the presence of inducing factors [220]. It is interesting how single chains are able to bind to this receptor. As the length of the sugar chain increases, more binding sites areas are present, making it less likely that the HA polymer would dissociate from the cell surface. The minimum binding chain length is 6 sugar residues. As the molecular weight increases, the binding avidity increases [108].

RHAMM is a receptor for hyaluronan-mediated motility, known also as CD168, and it was the first cell-associated hyaladherin discovered and described [221,222]. This receptor, encoded by the *HMMR* gene (in humans on chromosome 5, in mice on 11) [223], is present on the cell surface but also intracellularly in the cytoplasm and nucleus [224]. RHAMM protein lacks a transmembrane domain but is captured on the cell membrane by a glycosylphosphatidylinositol-anchor [225]. In the body, there is also a full-length protein (85 kDa) associated mainly with the interphase mitotic spindle (also termed IHABP) and three more isoforms may be generated by alternative exon splicing [226,227]. The biological function of RHAMM is strictly determined by its expression and localization in the cell [228]. Binding of exogenous HA to RHAMM on the cell surface plays a key role in the activation of signaling cascades, especially of protein tyrosine kinase, extracellular signal-regulated kinase (ERK) [229], focal adhesion kinase (FAK) [230], and Src [231] in mutual cooperation with CD44 [232,233]. Studies with CD44^−/−^ knockout mice indicate that RHAMM is able efficiently to promote inflammatory signaling due to the increased accumulation of HA by an HA-dependent mechanism [234]. In contrast, intracellular RHAMM binds to actin filaments and both interphase and mitotic spindle microtubules [235,236].

The RHAMM/HA interaction in cytoplasm strongly suggests that intracellular HA can influence the mitotic spindle and directly or indirectly modulate mediated signaling activity [237,238]. Telmer et al. claim that intracellular RHAMM can bind directly to MEK1/ERK1,2 and have proposed a model in which RHAMM functions as a scaffold protein [239]. Moreover, RHAMM expression increased during neoplastic progression in a variety of human tumors, mainly in breast cancer with poor prognosis, pancreatic tumor [240], multiple myeloma [241], fibrosarcoma [242,243], or colorectal cancer [244].

Another protein identified and homologous to the CD44 receptor, LYVE-1, was expressed almost exclusively on lymphatic endothelium [163]. This lymphatic vessel endothelial receptor is an extracellular, transmembrane protein that binds HA via a conserved Link domain [245]. In the human organism, it is encoded by the LYVE1 gene, which also is located on chromosome 11. LYVE-1 is expressed on the lymphatic endothelium, where it is involved in regulation of tissue fluid homeostasis, immune surveillance, and cell trafficking in lymphatic vessels and nodules [246]. It has also been detected in sinusoidal endothelial cells within the spleen and liver and in reticular cells of lymph nodes [247,248]. This receptor’s presence also has been confirmed on the endothelium of lung, adrenal gland, and heart, but it is absolutely not present on the endothelium of blood vessels [245,249]. The expression of LYVE-1 by a subpopulation of tissue macrophages also has been confirmed [250,251]. LYVE-1 is the main receptor responsible for the transport of HA from the tissues and degradation within lymph nodes, and it is an important molecular marker in the studies of normal and pathological lymphangiogenesis [252]. Physiologically, LYVE-1 begins to be expressed during embryonic development, almost simultaneously with the PROX-1 protein, but it has been shown later not to be necessary for normal lymphatic development and function [253,254]. The sugar polymer length required for optimal LYVE-1/HA binding is ≥22 sugars for maximal interaction, and the minimum bound oligomer contained eight units [255]. Despite strong homology between CD44 and LYVE-1, they differ in substrate specificity. Despite strong homology between CD44 and LYVE-1, they differ in substrate specificity. LYVE-1 shows exclusivity for HA binding and no affinity for other GAGs, but it can serve as a high-affinity receptor for growth factors like platelet-derived growth factor (PDGF) and vascular endothelial growth factor (VEGF) [256]. The LYVE-1/HA binding mediates docking of leukocytes and their entry into lymphatic vessels, as well as dendritic cell and macrophage trafficking. LYVE-1 is exploited by members of *Streptococcus group* A, and their dissemination to lymph nodes via their HA capsule causes post-infection sequelae [177].

The second receptor identified is HARE, and it, too, is involved in clearing HA from the circulation. It is an endocytosis receptor but also contains a binding module of the Link superfamily and is specifically expressed on sinusoidal endothelium of liver, spleen, and lymph nodes (together with LYVE-1) and is not found in lymphatic capillaries [164]. In humans it is encoded by the STAB2 gene, found on chromosome 12, and two isoforms are present of 190 and 315 kDa. Both HARE isoforms are functional endocytosis receptors via a coated pit and with identical ligand binding [257]. HARE is a promiscuous catcher of glycosaminoglycans, responsible for the systemic clearance of 14 different ligands, including not only HA but also four different chondroitin sulfates, dermatan sulfate, heparin, as well as non-glycosaminoglycan ligands such as acetylated low-density lipoprotein, pro-collagen pro-peptides, or glycation end products [258]. The binding of HA to HARE is size dependent. The length of a binding-capable fragment ranges from 2 kDa to 10 MDa (8 to 50,000 sugars). Any HA molecule with more than 8 sugars is capable of reaching the binding site of the domain, can bind to the HARE, and can be internalized. With growing chain length, the binding affinity also increases [259]. Binding affinity has been proven for HA fragments 40–400 kDa in length. Maximal signaling response occurs with HA of 107 kDa. Neither larger (436, 549, 967 kDa) nor smaller (14 kDa) HA can stimulate cellular NF-κB signaling through HA/HARE binding [260]. This transcription factor is ubiquitously expressed and plays an important role in the regulation of many genes encoding pro-inflammatory cytokines, chemokines, growth factors, and adhesion molecules. Also, HA/HARE binding, similarly to that of RHAMM, leads to activation of MAPK/ ERK1/2 in a concentration- and time-dependent manner, but the HARE protein alone is capable of forming complexes also with other MAP kinases, with c-Jun N-terminal protein kinase (JNK), and with p38 [261]. We next will mention a group of Toll-like receptors (TLRs) that HA also has been shown to activate. TLRs belong to the group of type I transmembrane proteins with ectodomains that mediate the recognition of pathogen-associated molecular patterns (PAMPs). They occur on the surface of cytoplasmic membranes (TLR1, 2, 4–6, 11). They are primarily expressed on antigen-presenting cells, such as macrophages and DCs, but also on epithelial cells of the intestine, kidney, lung, and cornea or intracellularly on vesicles (TLR3, 7–9) [262]. TLRs play an indispensable role in the body because they mediate the first line of defense against exogenous and endogenous PAMPs and activate innate and adaptive immune responses. TLR4 and TLR2 are the major receptors recognizing bacterial cell wall components. TLR4 is a receptor for lipopolysaccharide (LPS), and TLR2 mediates cellular responses to lipoproteins, lipopeptides, and lipoteichoic acid from Gram-positive bacteria, fungi, parasites, and viruses [263]. Generally, TLRs mediate signaling after binding of adaptor proteins, and the major adaptor for all TLRs (except for TLR3) is myeloid differentiation factor (MyD) 88. After binding, NF-κB, which acts as the inflammation master switch, is activated. Subsequently, MAP kinases, p38, and JNK are activated, and these also are involved in the increased transcription of NF-κB. Activation of NF-κB signaling can induce strong production of the inflammatory cytokines interferon (IFN) type I, interleukin 6 (IL-6), and tumor necrosis factor α (TNF-α), which subsequently activate surrounding cells to produce additional chemokines and adhesion molecules, thereby recruiting distinct cells to the inflammation site [264]. The work of Schibner et al. [129] demonstrates that LMW-HAs (200 kDa) activate inflammatory gene expression through TLR-2 in MyD88-dependent manners equivalent to LPS stimulation. On the other hand, intact HMW-HAs can inhibit TLR-2 signaling [129] and application of lovastatin, a cholesterol-lowering statin, inhibits LMW-HA-induced inflammatory reaction [265]. Similar results are presented also in the study of Campo et al. [131], who show that an LMW-HA (50 kD) enhanced expression of TLR-4, MyD88 and another adaptor protein, TNF receptor-associated factor 6 (TRAF-6), NF-kB activation, and expression of inflammation mediators. MMW-HA (1000 kD) showed no significant effect on NF-kB activity, but HMW-HA (5000 kD) significantly reduced NF-kB activation in LPS-stimulated cells [131]. Those authors also document a mechanism of action similar to that of LMW-HA for a tetraoligomeric HA fragment [266]. Moreover, in vivo experiments have confirmed that HA fragment (units of 4–16 oligosaccharides) is involved in activation of DCs by TLR4-dependent mechanisms [130]. A different mechanism of action, however, has been demonstrated in primary mesangial cells. In these cells, the applied 3 kDa HA did not influence TLR signaling directly but did affect the accessibility to TLRs by production of a jelly-like barrier on the cell surface [267]. A direct link was proven also between the interaction of oligo-HA (3–5 kDa) and CD44 and TLR2/TLR4 association with the actin filament-associated protein AFAP-110. This complex plays a key role in signaling that leads to stimulation of tumor invasion. The complex has been shown to mediate and produce critical components in the tumorigenesis pathway in breast tumor cells [268].

Currently, more than 35 hyaladherins are known, and new proteins that will be shown to bind or interact with HA are continuously being identified. Examples include cell migration-inducing protein (CEMIP), also known as KIAA1199 or HYBID [269]. CEMIP is a protein of emerging interest, the exact functions of which are still unknown. In addition to its key role in HA catabolism, it seems to be an early actor in turnover of what can be termed “chondro-myo-fibroblasts” and is responsible for ECM remodeling by promoting pro-fibrotic markers [270]. Also, identification of hyaluronan and proteoglycan Link proteins (HAPLNs) 1–4 brought novel insights regarding the organization of the HA-dependent ECM function [181]. Moreover, even proteins that have been identified in the past are being found to have new biological properties and functions. For example, HA binding protein 2 (HABP2), also known as factor VII activating protease, is an extracellular serine protease found in plasma. This protein contains the polyanion binding domain (PABD) created by two epidermal growth-factor-like (EGF) domains [271]. A differential ability of this protein to interact with HA was discovered. LMW-HAs increase the expression of HABP2, but, surprisingly, HMW-HAs behave in an opposite manner, inhibiting protease activity [272]. In addition to its involvement in pathological processes in several diseases, including atherosclerosis and deep venous thrombosis, a direct connection was found between the expression of HABP2 and an important factor contributing to alveolar homeostatic balance in the lungs [273], as well as, most recently, an important regulator of lung cancer’s progression [274]. Despite all the progress in discovering and describing the molecular mechanisms of hyaladherins and the implications of HA/HABPs binding, we continue to find many gaps remaining to be filled. In the future, the ongoing research on this issue will definitely provide answers important for improving human health and influencing cancer cell growth, invasion, metastasis, and other pathologies.

## 5. New Roles in Radiology and Biology

Exposure to ionizing radiation (IR) is part of our daily life. Although IR sources are primarily of natural origin, coming from the environment, a not insignificant percentage of such exposure is from medical irradiation. Despite its undeniable benefits, any diagnostic or therapeutic radiation exposure carries some risks. As many as 50% of all cancer patients undergo radiotherapy [275]. Despite huge advances in radiation oncology and those new methods have been developed to better target that radiation, better adapt it to the shape of the tumor, and reduce the applied dose in order to limit the amount of radiation absorbed by the surrounding tissues, we are still not able absolutely to eliminate the adverse side effects of IR [276]. The biological effect of IR after absorption of the dose by living cells consists in direct breaking of chemical bonds in biological macromolecules, such as proteins, complex lipids, and especially the DNA in the cell nucleus (accounting for 30–40% of damage). Most damage to DNA is caused indirectly, however, namely by the radiolysis of water by high-energy photons. Mechanism of water radiolysis includes 3 stages (physical, physico-chemical and chemical) differing in time scale. Physical stage consists energy deposition–ionization of matter from high-energy photons and subsequently relaxation processes. Water, as the most abundantly occurring molecule in the organism, is the most likely target. This results in production of ionized (H_2_O^+^) and excited (H_2_O*) water molecules [277,278]. Physico-chemical stage of radiolysis leads to the production of hydroxyl radical (OH•), hydrogen radical (H•), hydrogen peroxide H_2_O_2_, and free electrons e−aq. In the presence of oxygen, e−aq and H• atoms are rapidly converted to superoxide (•O_2_^−^) and perhydroxyl (HO_2_•) radicals, generally known as reactive oxygen species (ROS) [265,266]. These ROS react with surrounding molecules (chemical stage). The most significant effect is on DNA molecule, due to ROS produce two major forms of DNA damage: double-strand breaks (DSBs) (the most lethal form of damage) and base lesions (which are repaired by the base excision repair pathway). The resulting damage manifests as early cell damage and/or late damage [279]. Acute damage occurs due to repair failure and death of resident epithelial, endothelial, and immune cells, the result of which is activation of DNA damage response (DDR). Long-term harmful effects include damage to specific tissues and organs within the radiation field or genomic instability, which can result in an accumulation of mutations and carcinogenesis [280]. For this reason, understanding radiobiology is of paramount importance. Decades of research into the consequences of IR for DNA damage mechanisms is bearing fruit. Over the past three decades, significant progress in radiation research has advanced and substantially improved existing knowledge in clinical radiotherapy and in drug- and gene-targeted therapeutics applications [281]. These new paradigms clearly include the discovery of non-targeted and delayed radiation effects during the 1990s (nicely summarized by Mothersill [282]) and cell damage due to radiation-induced bystander effects (summarized by Mukherjee and Chakraborty [283]). Because the ECM and its composition are significantly affected by both direct and non-targeted effects of IR, HA, as a major component of the ECM, will also contribute to the resulting signaling of those processes induced. In the following paragraphs, we will describe the importance of HA and its metabolism in response to radiation exposure and the related injuries. 

As mentioned previously, the greatest danger is posed by ROS that are formed after the organism’s exposure to IZ. Reaction between ROS and HA plays an important role in the organism, but the attack on the HA backbone is a rather complex hydrolytic reaction. As mentioned earlier, degradation of HA by ROS constitutes one of its metabolic pathways for degradation in the body. Free radicals randomly cleave side groups from HA chains, dramatically affecting the tertiary structure, and after a longer exposure there occurs fragmentation with a consequent increase in the number of small HA oligosaccharides. Hydroxyl radicals are most effective in breaking glycosidic bonds in HA. The cross-reaction results in breakage of H• and the formation of a carbon-centered radical, with each radical formed having a specific individual chain. The efficiency of C–H bond breakage is nearly 100% and results in products with different chain lengths ranging even to monosaccharides [284,285].

Detecting the effect of IR on HA after exposure, either of the molecule itself or in solution, has been a challenge since its discovery. As early as 1950, a significant change in the viscoelastic properties of HA was found that is directly dependent upon the intensity of radiation and chain depolymerization by X-rays [286,287]. The irradiation of HA (6MV X-ray, range 0–20 Gy) ionizes and excites the HA atoms and the surrounding ECM to such an extent that can lead to changes in the physical and chemical nature of the polymeric HA. Chain scission results in a decrease in molecular weight and associated viscosity [288]. In their work, Al-Assaf et al. [289] found HA chain damage caused by radiolysis-generated hydroxyl radical to be at the level of 52% and 44% of breaks in the absence and presence of oxygen conditions, respectively. This work also showed the interaction of HA with IR (10 TBq, 137Cs gamma source) radicals leading to chain scission to be a random process and that, with increasing IR doses, the number of breaks was greater and so was the decrease in HA’s molar mass. Similar results have been reported from several other studies using neither 60Co or 137Cs gamma sources [290,291]. Another study by Ahmad et al. [292] investigated the effect of gamma irradiation (60Co, doses 0–200 Gy) on HA and articular phospholipids. They found that HA is fast depolymerized, and the glycosidic bound is cleaved with the formation of carboxylic acid (C(=O)OH). A novel application was hypothesized by the team of Huang et al. [293]. They analyzed physical and chemical properties of LMW-HA formatted from irradiated HMW-HA (60Co gamma source, doses 20–60 kGy). Shorter fragments were created by increasing IR doses in a dose-dependent manner; nonetheless, because the solid powder was irradiated, they showed that LMW-HA fragments were more homogenous compared with the irradiation of the HA liquid solution. They increased the number of the carbonyl group, but without significant alternations in fundamental structure [293]. Despite the fact that these results were not completely consistent across studies due to different experimental setups and different methods for analyzing the resulting HA fragments, which can cause there to be significant differences in chain yields. This problem is even more complex under in vivo conditions, where the size of HA fragments is an important determinant of individual biological properties and activated processes.

ROSs, however, are naturally produced in the body from mitochondrial metabolism, and under physiological conditions, they serve as essential signaling molecules that regulate numerous cellular processes. Other cellular sources of ROS, and primarily of the •O_2_^−^ radical, are neutrophils, monocytes, macrophages, and eosinophils when increased concentrations occur through the action of NADPH oxidase during a so-called “respiratory burst”. This process, one of the first lines of defense against environmental pathogens, is a part of phagocytosis [294]. NADPH oxidase is a highly regulated enzyme complex that reduces oxygen to •O_2_^−^ in vivo. The final product of the degradation is the OH• radical produced by the Fenton reaction in the presence of low valent transition metals (Fe^2+^ or Cu^2+^) [295]. Two other enzymes play important roles in this enzyme complex, namely myeloperoxidase (MPO, EC 1.11.2.2), which produces hypochlorous acid (HClO) from H_2_O_2_ during a neutrophil’s respiratory burst, and superoxide dismutase (SOD, EC 1.15.1.1), which converts superoxide •O_2_^−^ into hydrogen peroxide that can be further converted into water and free oxygen by a group of catalase enzymes. Under pathological conditions, an increased amount of ROS leading to oxidative stress and cytotoxicity causes a loss of cellular functions and the development of heterogeneous disease such as inflammation or cancer [296]. The spectrum of ROS generated during and shortly after irradiation is similar to that produced by metabolic processes. There are differences, however, in microdistribution, the relative yields of specific products (mainly •O_2_^−^ and H_2_O_2_ produced by endogenous processes vs. the highest OH• yields after irradiation), and the timing of the production (chronic release of endogenous ROS versus immediate production during irradiation) [297]. In addition to IR, HA can be degraded also by UV light [298], heat [299]; antioxidants such as ascorbic acid [300] and copper ions; and ultrasound [301]. Reactive nitrogen species (RNSs) also participate in HA degradation. RNSs are created in the body similarly as are ROS, but the starting molecule is nitric oxide (NO). The most reactive species attacking HA are peroxynitrite anion (ONOO^−^) and HClO generated by myeloperoxidase [302]. When reacting with ONOO^−^, a decrease in molecular weight and increase in polydispersity have been observed, thus suggesting a progress and mechanism similar to those of OH• [303,304]. Based upon the information described above, it is clear that HA’s reaction with IR is rapid and very intense. This is not exactly beneficial, however, because there is increased accumulation in the body of unwanted short fragments. From this point of view, it appears that HA will not have a prominent place in radiation biology but rather the opposite, and it will contribute to the damage and, moreover, promote the induction of an inflammatory response induced by IR. However, is this really the case?

A clear and direct answer was provided by the work of Riehl et al. [305], whereby the individual molecular mechanisms that clearly confirm the radioprotective effect of HA were successfully identified and explained. One of the main goals of radiobiology is to identify new and better compounds that reduce radiation toxicity. Radioprotection concerns prevention and mitigation of radiation-induced damage. Radioprotective effects are mediated by exogenous HA given before irradiation through binding to TLR4 and cyclooxygenase-2 (COX-2). In the excellent review by Ratikan et al. [306], the concepts of TLRs and IR danger signaling are examined and summarized in great detail. COX-2 is an enzyme that involves prostaglandin E2 (PGE2) synthesis. Prostaglandin-induced radiation protection was reported as early as 1972 [307]. Administration of intraperitoneal HA 8 h prior to irradiation has radioprotective effect mediated through induction of COX-2 and PGE2 gene expression in the intestine, increased migration of mesenchymal stem cells from villus to crypt, and reduction in radiation-induced apoptosis [305]. This study, having done a tremendous amount of laboratory work and brilliant summarizing the existing knowledge, thus opens up a whole new perspective on HA as a possible clinically useful radioprotective agent. The gastrointestinal syndrome is pathological due to the lack of cells replacement in the surface of the villi because stem and proliferating cells located in the crypts of Lieberkühn are damaged. It is one of the forms of acute radiation syndrome (ARS), manifesting a pattern of physiological responses in the most radiation-sensitive organ. Otherwise, the preferred model today for describing subsequent radiation concomitant and interdependent injuries to various organ systems is that of radiation-induced multiple organ dysfunction/failure syndrome (RI-MODS/RI-MOF) [308,309]. Regardless of the terminology, there is still no effective treatment for this syndrome and each new study can improve the prospects for radiation therapy patients as well as for victims of nuclear accidents. 

The lung is another organ within which HA plays a very important role in the course of radiation-induced damage (Figure 3). HA is the major nonsulfated glycosaminoglycan in the lung. The importance of its signaling in lung tissue has been clearly described under physiological conditions but also in relation to various pathological events [310,311,312]. In this review, however, we will focus solely on defining the role of HA during radiation response. In addition to HA, also its individual fragments, hyaladherins, and ECM are important factors in regulating IR damage to lung. Pulmonary irradiation can produce a great deal of ROS which cause severe injury or apoptosis of alveolar epithelial cells and vascular endothelial cells causing extensive production of large amounts of inflammatory cytokines, chemokines, and growth factors with resolution of inflammatory reactions and chemotaxis of monocytes, lymphocytes, and granulocytes that gather at the injured lung tissue. Radiation-induced pulmonary injury (RIPI) may result when a dose absorbed by the tissue prevents repair mechanisms or causes low efficiency in repopulating to replace damaged and/or dead lung tissue. Subsequently, two undesirable pathological syndromes develop over time. The first is radiation pneumonitis (RP), as an acute reaction within 4–12 weeks. The second, but irreversible phase is radiation fibrosis (RF) occurring 6 months after irradiation. These two pathologies constitute the main limiting factors for efficient radiotherapy of the thorax region, and therefore new strategies which could improve therapeutic outcomes are still sought [313]. In the lung, HA is prominent among GAGs representing potential ECM targets for ROS. Studies have shown that oxidative stress plays an important role in modulating the ECM promotion of fibrosis in the lung [314]. The work of Gao et al. [315] pointed out the possible protection of HA fragmentation by the extracellular superoxide dismutase enzyme (EC-SOD, also referred to as SOD3). This enzyme is highly expressed in mammalian lungs and the mechanism of prevention against ROS degradation lies in the EC-SOD/hyaluronan interaction, because it has been proven that purified mouse EC-SOD binds directly to hyaluronan in vitro. This study also confirmed this protective mechanism under in vivo conditions. The mouse model was exposed to asbestos-induced lung inflammation and presence of EC-SOD inhibiting ROS-induced fragmentation of hyaluronan from the matrix to the airspace and plasma [315]. Similar findings as in this study were obtained also when using a model of bleomycin-induced acute lung injury [316]. Because oxidative damage is also a central pathogenic process in RIPI, amelioration of lung tissue by SOD3 was proven [317,318,319]. Because to date there has not yet been a direct study unambiguously describing the EC-SOD/hyaluronan interaction in the RIPI model, new research opportunities are opening up.

The dynamics of changes in HA concentrations after partial irradiation in lung tissue, bronchoalveolar lavage, and serum are well mapped [320,321]. A study by Li et al. [322] confirmed that after irradiation of the lower part of rats’ right lungs, HA accumulated in bronchoalveolar lavage fluid (reaching its highest level 6 weeks after irradiation). Also, the expression of genes associated with the HA signaling pathway was significantly altered after irradiation. The HAS2 gene expression was increased 4, 6, and 10 weeks after irradiation, while expression of both HYAL1 and HYAL2 peaked at 4 weeks. This study also confirmed that tissue fibrotic remodeling factors TGF-β1 and PDGF affect HAS2 gene expression and HA production. The investigators also found that HA and/or HA fragments could play an important role in lung tissue remodeling by activating HYALs and inducing types I and III collagen expression [322]. 

It is not only HA that responds to IR, however, but also individual components of its metabolism and its interacting partners respond. 

HAS2, the synthetic enzyme, has been studied most intensively in the context of IR and also of fibrosis. The work of Schen et al. confirmed that HAS2 knockdown contributes significantly to IR-induced DNA damage and radioresistance of cancer cells. The anticipated mechanism is through HA/CD44 interaction and activating the epidermal growth factor receptor (EGFR) signaling pathway [323]. Further, other cells, such as fibroblasts, can be stimulated by proinflammatory cytokines to produce ROS. Tissue hypoxia resulting from vascular damage is another continual source of ROS generation. Several enzymes are now recognized as being potentially able to produce ROS; perhaps the most important of these is NADPH oxidase. Another source of ROS is xanthine oxidase. The generation of these reactive molecules is part of the innate immune system and helps to rapidly cleanse the wound of injury, but excessive production of ROS can lead to severe tissue damage, including fibrosis and even neoplastic transformation [324]. A study by Li et al. [325] provides evidence about a connection between severe RF and hyaluronan signaling pathway. Overexpression of HAS2 on fibroblast isolated from transgenic mice was connected with invasive fibroblast phenotype. Their study confirmed a fundamental role for HAS2 in the development of RF. Furthermore, treatment of IPF fibroblasts with anti-CD44 antibody markedly reduced invasive capacity, emphasizing the contribution of CD44 in progressive fibrotic phenotype. Based upon these results, those authors present the idea that severe RF phenotype requires contact with the matrix [325]. 

Another signaling pathway that has been identified as supporting the development of pulmonary fibrosis after irradiation is the CD73/adenosine interaction. CD73 is ecto-5′-nucleotidase (also known as NT5E), an enzyme generating extracellular adenosine [326]. The contribution of CD73/adenosine signaling to RF pathogenesis is well known [327,328]. Thoracic irradiation-triggered upregulation of CD73 in lung tissue leads to extensive generation of extracellular adenosine promoting lung tissue remodeling and progression of chronic lung inflammation to fibrosis. Also, environmental hypoxia activates molecular mechanisms that result in increased CD37 expression and elevations of adenosine. Moreover, several clinical studies have been conducted on pharmacological modulation of adenosine levels to limit lung toxicity during radiotherapy [327]. A new perspective, however, was introduced by the work of de Leve et al. [329]. In this work, CD73/adenosine signaling was shown to be important for the generation of alveolar macrophages with alternatively activated phenotype that accumulate in organized clusters within lung tissue and express fibrotic marker proteins. The aforementioned molecular mechanisms are important processes in the induction of RF. Interestingly, CD73 deficiency was shown significantly to affect the HA signaling pathway in irradiated lung tissue. The main manifestation was reduced HA deposition in lung tissue and differential expression of individual Has genes. These studies confirm the association between adenosine and HA pathways in the regulation of tissue inflammation and fibrosis in mouse models [329]. 

Furthermore, expression of RHAMM is downregulated by tumor suppressor p53, the so-called “protector of genomic stability.” P53 is responsible for induction of cell-cycle arrest, apoptosis, and depression of DNA repair after radiation. It has been shown that p53 can repress RHAMM expression via suppression of its transcription. IR activates p53-mediated signaling in a variety of cells, but the consequence of p53 activation is cell-type dependent. In lung tissue, p53 participates in vascular homeostasis and protective inflammatory reactions. During fibrotic processes, p53 levels are increased in lung tissues and epithelial cells. RHAMM is an important regulator of proper cell cycle progression, and suppression of RHAMM may be an oncogenic feature in lung tissue [330,331,332,333]. Many knowledge gaps remain to be filled in order to understand completely the potential of HA and its proteins in the field of radiation biology.

## 6. That Is Not All

Based upon what has been detailed above, we can say that HA has potential for application also in the fields of radiobiology and radiation oncology. Exact mechanisms for its possible practical applications, however, remain to be elucidated. Due to tremendous progress in studying HA biology, this polymer is being examined intensively also across other fields where it may find applications (Figure 4). Therefore, we will briefly mention a few more of these.

The use of hyaluronic acid in the pharmaceutical and cosmetic industries is well documented, from its general properties, such as lubrication and space filling, to more specific characteristics, such as immunosuppression and tissue recovery. Besides these positive effects, the natural degradation of hyaluronic acid allows it repeated application. In ocular therapy, for example, hyaluronic acid is tradionally used in combination with other polymers that can delay its degradation [334,335,336]. A long-established therapeutical application includes the use of hyaluronic acid in osteoathritis and other joint injuries; it must be mentioned that in these cases only its high-molecular weight form is applied. The low-molecular weight form of hyaluronic acid, or its fragments, could be used through oral administration because they are more easily absorbed [337,338]. The popularity of hyaluronic acid has increased yearly, not only in the research environment but also among the general population. Because of its viscoelastic, space-filling and hydrophilic properties, hyaluronic acid is used in many everyday cosmetic products to retain skin moisture or eye lubrication. Eye drops and nose sprays are among the products with additional hyaluronic acid content that can be used without a prescription [339,340,341]. Hyaluronic acid has become popular in professional cosmetic studios and plastic surgery, where it is applied as a filler to reduce wrinkles and skin rejuvenation, where it can be applied on its own or in combination with botulotoxin A or laser treatment [342,343,344,345,346,347].

Hyaluronic acid and its metabolic pathway and receptors have always been intensively studied in the context of tumorigenesis. Correlation between the expression of particular variants of some hyaladherins and individual tumor types has already been mentioned in the previous section, but in studying short HA fragments, an interesting association with breast cancer has been found. Wu et al. [348] were the first to point out that serum levels of LMW-HA (smaller than 50 kDa) but not the total HA level in patients correlate significantly with breast cancer metastasis. In the future, this might mean that measuring LMW-HA levels in serum could serve as an effective screening tool for metastasis in this type of cancer. Also, the authors of this study have pointed out the possibility for therapeutically influencing breast cancer by modulating the HA metabolic cycle.

A major breakthrough occurred in the early 1990s due to the successful covalent linking of HA with cytotoxic agents. This new perspective drew enormous attention to HA as a biomaterial suitable for drug conjugates. The chemical coupling of a cytotoxic drug to HA improves the pharmacokinetic profile of the drug, prolongs drug distribution, and reduces the drug therapeutic window. Drug delivery applications include anti-inflammatories, antirheumatic antibiotics, analgesics, and even DNA or siRNA molecules for gene therapy. Targeted delivery due to the HA/CD44 interaction is an advanced option for promoting tumor drug delivery. The HA anticancer conjugates include sodium butyrate, cisplatin, taxols, doxorubicin, 5-fluorouracil, and mitomycin C. This delivery approach has higher therapeutic efficacy. Recently, HA has been the most adaptable biopolymer used in tissue regeneration and medicine. The most effective application involves its use as scaffolding for cartilage repair, cardiovascular tissue engineering, or as hydrogels for brain and neural regeneration, as well as for wound healing and skin regeneration [349,350,351].

Already in the 1970s, Margnolis et al. [352] and Stein et al. [353] put forward the idea that HA could occur also intracellularly. Later, in the 1990s, the presence of intracellular HA was confirmed in the nuclei of liver cells, rough endoplasmic reticulum, and Golgi apparatus. To date, two main molecular mechanisms are thought to be responsible for intracellular localization of HA: endocytosis from the pericellular matrix via binding to a specific receptor, or synthesis by cytoplasmic HAS after their endocytosis from plasma during degradation or by newly synthesized and activated HAS directly in the cytoplasm or localized on a specific intracellular component [354]. The exact mechanisms of why and how HA is synthesized inside the cytoplasm and HAS are activated remain obscure. Nevertheless, we do already know that intracellular HA plays an important role during inflammatory processes [355] and that in hyperglycemic conditions active HAS enzymes produce intracellular hyaluronan, which is implicated in the pathogenesis of diabetes [356,357].

Last, but definitely not least, we should highlight the role of HA in nanoscience and nanomedicine. Hyaluronic acid enhances the hydrophilic properties of other molecules; however, it could also be used to eliminate cytotoxicity, e.g., quantum dots. Quantum dots, semiconductor particles in nanosize, are extensively used as a tool in imaging implementation tests because of their easily controlled optical properties. On the other hand, the toxicity of traditionally used heavy metals, such as cadmium, limits its utilization in vivo [358,359]. This feature could be eliminated by surface modifications with amphiphilic polymers or hyaluronic acid. Further, its functional groups could be used in other modifications [360,361]. Currently, the potential of hyaluronic nanoparticles in imaging techniques is being studied, mostly because of its ability to bind to tissues overexpressing CD44, as is the case of some cancer tumors. In this regard, hyaluronic acid is used as a backbone or as a hydrophilic shell for other hydrophobic cores. Near-infrared labels conjugated with Cy5.5 have also been used in the past, as well as other conjugates based on luminescent semiconductor nanocrystals or gold nanoparticles [24,362,363]. Another approach relies on direct crosslinking of hyaluronic acid with diethylenetriaminepentaacetic acid as a chelating reagent for gadolinium ions [364]. Another contrasting agent, cetyltrimethylammonium-bromide, has been used for the subsequent functionalization of hyaluronic acid nanoparticles [365]. All of the mentioned approaches have been applied in magnetic resonance imaging techniques. Due to the promising chemical structure of the HA molecule and its hydrophilic backbone, it can easily be modified chemically to produce HA nanoparticle (HANP) structures [366]. This area of research is developing very rapidly. Our team has successfully developed a methodology for highly efficient enhanced HA fragmentation in order easily and safely to produce molar mass-defined HA fragments and form HANP structures [367]. It also has used HANP for therapeutic purposes in treating RIPI. We have observed significant effects on molecular and cellular patterns during the radiation fibrosis phase [368]. HANP conjugates with the antitumor drug doxorubicin are suitable for treatment of even aggressive brain glioblastomas and produce better therapeutic outcomes [369]. A few HA-based nanomaterials already have been investigated successfully in clinical trials. This progress underscores the tremendous potential this molecule can have in targeted delivery and therapy of numerous diseases [370,371].

## 7. Conclusions

Since its discovery, hyaluronic acid has remained one of the most analyzed, studied, and functionally modified molecules of our time due to its unique hydrophilicity, viscoelasticity, non-immunogenicity, biocompatibility, and degradability. Its derivatives are currently used in many different biomedical applications, including arthritis treatment, ophthalmology, tissue engineering, and wound healing, and they are being tested for use in more efficient drug-delivery systems in various advanced forms such as hydrogels or nanoparticles. Despite continuous scientific advances, this simple molecule is forever finding ways to surprise us with its many potential uses, even though its exact mechanisms of action in wound healing and tumor biology, the origin and function of its very long chains during ontogeny, and the functions of intracellular HA and its associated partners are not yet known. In the future, our team will focus more on the relationship between HA and ROS elimination, the role of HA in inflammation regulation and radioprotectivity, and tissue damage/regeneration after irradiation as possible methods of applications. For these reasons, it can be anticipated that HA will remain in the spotlight for several more decades to come and that its research will not go out of fashion any time soon.

## Figures and Tables

**Figure 1 pharmaceutics-14-00838-f001:**
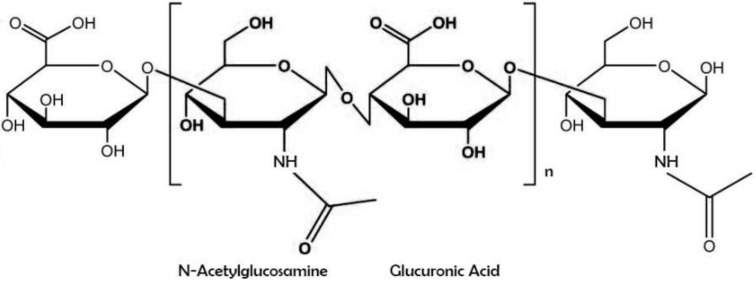
Chemical structure of hyaluronic acid with repeating disaccharide units of (β, 1–4)-glucuronic acid (GlcUA) and (β, 1–3)-N-acetyl glucosamine (GlcNAc).

**Figure 2 pharmaceutics-14-00838-f002:**
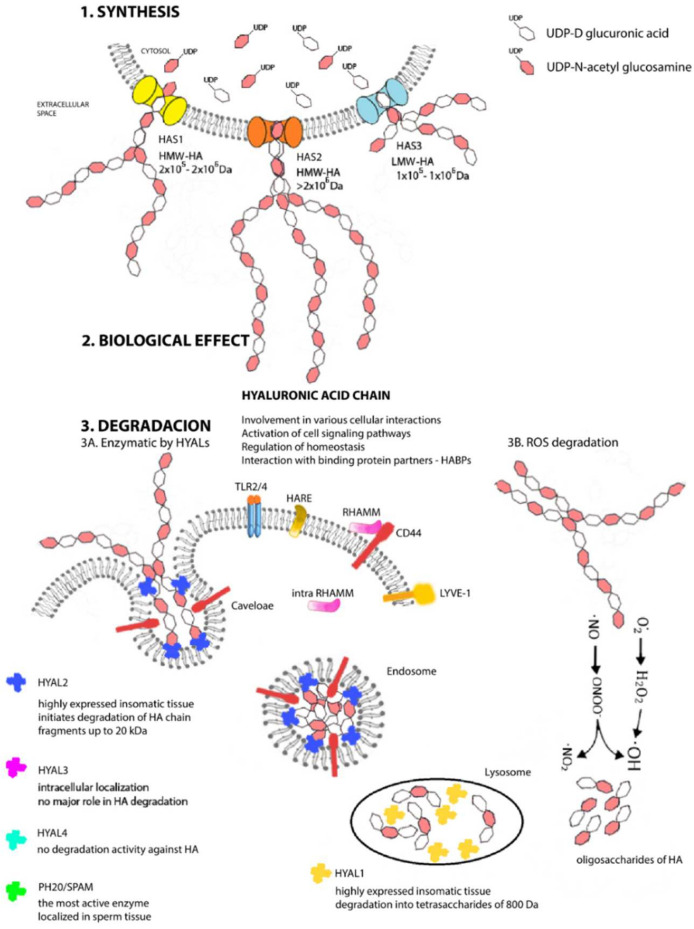
Metabolism of hyaluronan. Structural scheme of synthesis (**1**), biological effect (**2**), and degradation (**3**) of HA under physiological conditions in mammal cells. Cytosolic molecules of UDP-D glucuronic acid and UDP-N-acetyl glucosamine serve as precursors for HA chains. The enzymatic reaction is catalyzed by three HAS proteins, which synthetize unique HA chains varied in lengths that differ in their biological function in the organism (**1**). Hyaluronic acid chains are involved in many cellular interactions, in signaling pathways, binding activities with other proteins, or are involved in process of homeostasis (**2**). In vivo degradation proceeds in parallel in two ways (enzymatically (**3A**) and chemically (**3B**)). Specific hyaluronidases (HYAL1—HYAL3 and PH20/SPAM) are localized in different tissue of organism and initiates the degradation of HA chain (HYAL1 in lysosomes, HYAL2 in inner part of endosome membranes, and HYAL3 in cellular membrane). Enzyme PH20 SPAM is localized in testis. Free radicals provide the random chemical degradation (**3B**) of HA chains and after a longer exposure, fragmentations occur with a consequent increase in the number of small HA oligosaccharides.

**Figure 3 pharmaceutics-14-00838-f003:**
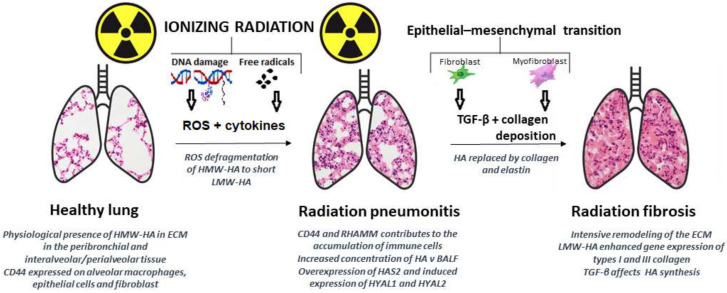
Radiation induces lung injury and hyaluronic acid. The mechanism of radiation pneumonitis is initiated by ionizing radiation, which causes extensive DNA damage and large-scale production of free radicals in lung tissue. Both processes trigger persistent inflammation with all consequences leading to pathological changes, including immune cell infiltration, capillary permeability, and pulmonary edema. Untreated pneumonitis leads to serious radiation damage of the lungs, which causes irreversible radiation fibrosis characterized by an accumulation of extracellular matrix proteins. Hyaluronic acid, protein in its metabolism, and HA-binding proteins are significantly affected by ionizing radiation in lung tissue and have a distinct impact on RIPI progression.

**Figure 4 pharmaceutics-14-00838-f004:**
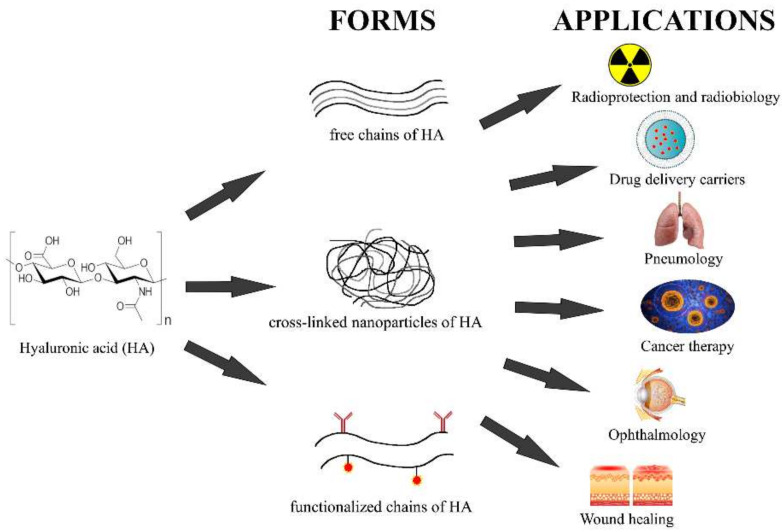
Applications of hyaluronic acid. Different forms of chemically modified HA—native, cross-linked or functionalized are used in pharmaceutical, medical, cosmetic, and research applications.

**Table 1 pharmaceutics-14-00838-t001:** Hyaluronan-binding proteins.

Localization	Domain	Binding Protein	Function	Reference
Cell-surface	Link module domain	CD44	regulation of cell–cell interaction and cell–matrix interface; mediation of cell adhesion; intracellular signaling pathways (Ras, MAPK, and PI3K)	[176]
		LYVE-1	lymphatic trafficking; hyaluronan degradation; intracellular signaling pathways for endothelial junctional retraction; regulation of lymphatic endothelial proliferation; lymphatic endothelium marker	[177]
		HARE/Stabilin-2	regulation of ligand binding and endocytic activity; mediation of hyaluronan clearance	[178]
		TSG-6	modulation of hyaluronan-CD44 interaction	[179]
		Stabilin-1	regulation of endocytic activity	[180]
		HAPLN 1–4	regulation of HA binding; HAPLN2 and HAPLN4 are specific for brain/CNS tissue	[181]
		Bral1	formation of the hyaluronan-associated matrix in the CNS	[182]
	B(X7)B motif	RHAMM/CD168	critical component of the inflammatory response	[146]
	Other	TLR2, TLR4	macrophage activation and proinflammatory response; stimulation of endothelial recognition	[183]
		ICAM-1	regulation of cell adhesion	[184]
		Layilin	mediation of cell adhesion	[185]
Extracellular	Link module domain	Hyalectins: versican, aggrecan, neurocan, brevican (BEHAB)	regulation of HA binding; forming aggregates with HA in ECM	[186]
		fibrinogen	regulation of HA binding in ECM	[187]
	B(X7)B motif	Trypsin inhibitor (IαI)	mediation of HA-TSG-6 binding	[188]
	Other	CEMIP (KIAA1199/HYBID)	included in cell-migration; hyaluronan depolymerization	[189]
		SPACR, SPACRCAN	protein in interphotoreceptor matrix in subretinal space; organization and support to photoreceptor function	[190,191]
Intracellular	B(X7)B motif	iRHAMM	cell division; binding to the mitotic spindle; interacting with microtubules and microfilaments	[192]
		USP17 (mouse SDS3)	regulator of cell proliferation and survival; essential for chemotaxis and chemokinesis	[193]
	Other	IHABP4	involved in cell interaction	[194]
		CDC37	cell division; essential cell cycle regulatory factor	[168]

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
