# Peer review of "Hyaluronic Acid: Known for Almost a Century, but Still in Vogue"

_pharmaceutics, 2022, doi:10.3390/pharmaceutics14040838_

Round 1

Reviewer 1 Report

The reviewer recommends this work can be published after a major review.The review “Hyaluronic Acid: Known for Almost a Century, but Still in 2 Vogue’’ needs to be address these aspects. 

  1. The authors should work on visulisation extensively. At least 4 informative figures need to be added to be more eyecathing.
  2. A part should be dedicated to other possible application of HA.
  3. There are a lot of reviews on HA, Therefore, the authors should comment on this and justify why there is room for another review.

Author Response

Dear Reviewer,

We have incorporated your suggestions to our revised manuscript.

We are grateful for your provided revision. 

Yours sincerely,

Anna Lierova

Reviewer 2 Report

The study, titled “Hyaluronic Acid: Known for Almost a Century, but Still in Vogue.” This review discusses the HA polymer may find applications in protecting against ionizing radiation (IR) or for therapy in cases of radiation-induced damage. I recommend publication of this review in Pharmaceutics after the minor revision.

Comments:

  1.    I find the organization with respect to the topic or theme or outline to be rather poor. The review would benefit from an outline in the introduction so that a reader could find their way to the subtopic of particular interest.
  2.    The title " Old and New Partners in Action." I suggest the authors emphasize the key issue HA degradation of tetra HA part.
  3. New Roles in RADIOlogy and Biology part explanation was limited. Suggest the authors emphasize the key issue of HA conjugation part with the application.
  4. The authors suggested the explain each section of figure 2.
  5. Authors must use a flow chart to explain each title discussion section. The reader will find it simple to read and understand.
  6. The author intends to describe the limitations of the other treatment methods in the introduction section.
  7. Finally, the dosage, timing, and duration period for IR sources remain unknown.
  8. Suggested to author revise the (1) abbreviation (2) superscript
  9. At Line 585 and 586 (reference no. 258), there is no detailed mechanism for water radiolysis by high-energy photons. Suggested the authors provide a more detailed description of the mechanism or references?

Author Response

Dear Reviewer,

We have incorporated your suggestions to our revised manuscript.

Thank you for your provided revision.  Should you need further

information, please do not hesitate to contact me at your convenience.

Yours sincerely,

Anna Lierova

Round 2

Reviewer 1 Report

The text in figure 4 should be improved, it is not readable.

Typically new parts and modification are highlited in a revision version, but I do not see any highlight.

Author Response

Dear Reviewer,

The new version with highlithed revision was submitted, as well as Figure 4 was edited.

Thank you for very fast responce.

Yours sincerely,

Anna Lierová